# The Acute Effects of a Fast-Food Meal Versus a Mediterranean Food Meal on the Autonomic Nervous System, Lung Function, and Airway Inflammation: A Randomized Crossover Trial

**DOI:** 10.3390/nu17040614

**Published:** 2025-02-08

**Authors:** Diana Silva, Francisca Castro Mendes, Vânia Stanzani, Rita Moreira, Mariana Pinto, Marília Beltrão, Oksana Sokhatska, Milton Severo, Patrícia Padrão, Vanessa Garcia-Larsen, Luís Delgado, André Moreira, Pedro Moreira

**Affiliations:** 1Basic and Clinical Immunology, Department of Pathology, Faculty of Medicine, University of Porto, 4200-319 Porto, Portugal; francisca_castromendes@hotmail.com (F.C.M.); vaniastanzani@gmail.com (V.S.); anaritamoreiraa@gmail.com (R.M.); mariana_cpinto@hotmail.com (M.P.); beltraomarilia@gmail.com (M.B.); o.sokhatska@gmail.com (O.S.); ldelgado@med.up.pt (L.D.); andremoreira@med.up.pt (A.M.); 2Serviço de Imunoalergologia, Unidade Local de Saúde de São João, 4202-451 Porto, Portugal; 3EPIUnit-Institute of Public Health, University of Porto, 4050-600 Porto, Portugal; milton@med.up.pt (M.S.); patriciapadrao@fcna.up.pt (P.P.); spcnaspcna@gmail.com (P.M.); 4Laboratory for Integrative and Translational Research in Population Health (ITR), 4050-600 Porto, Portugal; 5School of Medicine and Biomedical Sciences, University of Porto, 4050-321 Porto, Portugal; 6Faculty of Nutrition and Food Sciences, University of Porto, 4150-180 Porto, Portugal; 7Program in Human Nutrition, Department of International Health, Bloomberg School of Public Health, The Johns Hopkins University, Baltimore, MD 21205, USA; vanigarcialarsen@gmail.com; 8RISE-Health, Department of Pathology, Faculty of Medicine, University of Porto, 4200-319 Porto, Portugal

**Keywords:** airway inflammation, autonomic nervous system, lung function, fast food, mediterranean diet, pupillometry, randomized crossover trial

## Abstract

Background/Objectives: This study aimed to assess the acute effects of two isoenergetic but micronutrient-diverse meals—a Mediterranean-like meal (MdM) and a fast food-like meal (FFM)—on the autonomic nervous system (ANS), lung function, and airway inflammation response. Methods: Forty-six participants were enrolled in a randomized crossover clinical trial, consuming two isoenergetic meals: FFM (burger, fries, and sugar-sweetened drink) and MdM (vegetable soup, whole-wheat pasta, salad, olive oil, sardines, fruit, and water). Pupillometry assessed parasympathetic (MaxD, MinD, Con, ACV, MCV) and sympathetic (ADV, T75) nervous system outcomes. Lung function and airway inflammation were measured before and after each meal through spirometry and fractional exhaled nitric oxide (FeNO), respectively. Results: Mixed-effects model analysis showed that the MdM was associated with a hegemony of parasympathetic responses, with a significant increase of MaxD associated with a faster constriction velocity (ACV and MCV); on the other side, the FFM was associated with changes in the sympathetic response, showing a quicker redilation velocity (a decrease in T75). After adjusting for confounders, the mixed-effects models revealed that the FFM significantly decreased T75. Regarding lung function, a meal negatively impacted FVC (ae = −0.079, *p* < 0.001) and FEV1 (ae = −0.04, *p* = 0.017); however, FeNO increased, although after adjusting, no difference between meals was seen. Conclusions: Our study showed that the FFM counteracted the parasympathetic activity of a meal, while a meal, irrespective of the type, decreased lung function and increased airway inflammation.

## 1. Introduction

Diet plays a pivotal role in modulating airway inflammation [1], lung function [2,3], and the autonomic nervous system [4]. In particular, the Western diet, high in saturated fatty acids (SFA) and low in antioxidants, has been associated with increased inflammation and poorer lung outcomes in individuals with asthma [1]. Conversely, the Mediterranean diet, which is rich in unsaturated fatty acids, fiber, and antioxidants, has been demonstrated to reduce the prevalence and persistency of asthma [5], and adherence to this type of diet has been associated with improved lung function [6]. 

Beyond lung function, the autonomic nervous system (ANS) plays a crucial role in regulating the body’s response to both internal and external stimuli [7], interacting with immune responses through the gut–brain axis [8], stress regulation, fatigue, anxiety [9], and neuroendocrine functions [10]. Obesity [11] and environmental exposure [12,13] modulate ANS. Dietary habits [14] are also known to affect ANS function. Specifically, both ketogenic and vegetarian diets were shown to modulate ANS activity [15], while dietary patterns characterized by consuming sugary beverages modulated autonomic nervous system behavior [16]. 

Lung function and the ANS are closely interconnected, particularly in asthma [13,17]. The ANS regulates airway smooth muscle tone, bronchoconstriction, and mucus secretion. In this context, studies have shown that dietary or lifestyle interventions can modulate ANS responses and may impact respiratory outcomes. Specifically, long-term response to diet and exercise-based weight loss increased parasympathetic activity and decreased sympathetic activity, while the opposing effects were observed with weight gain [11]. In school-aged children, dietary diversity has been associated with decreased airway inflammation [18] and modulation of ANS responses [14]. Airway relaxation, mediated by the activation of β-receptors in the sympathetic pathway, may be influenced by dietary components such as omega-3 polyunsaturated fatty acids (PUFA) [19] or polyphenols [20]. By modulating ANS balance, dietary components and diversity might influence lung function. The ANS response to a meal may also vary according to physical activity level [21] and obesity [22].

Meal content induces different inflammatory responses; an MdM was shown to reduce postprandial ox-LDL, inflammation expression, and oxidative stress-related genes [23,24]. Also, the fat level in a meal was shown to increase airway inflammation [25], an effect that was counteracted by omega-3 supplementation. Nevertheless, the impact of a single meal has yet to be well studied in evaluating lung function, airway inflammation, and the ANS response. Therefore, we hypothesized that the content of a meal might induce different inflammatory responses that might be linked to lung function and the autonomic nervous system.

Given the potential of different diets to influence airway inflammation, lung function, and ANS, this study aimed to assess the acute effects of two isoenergetic but micronutrient-diverse meals—a Mediterranean-like meal (MdM) and a fast food-like meal (FFM)—on lung function and airway inflammation, but also on ANS response, as measured by pupillometry. Pupillometry, which is a simple and non-invasive method [26], has gained clinical relevance for assessing ANS function [27]. By measuring pupil dynamics, such as constriction and dilation, it reflects the balance between sympathetic and parasympathetic nervous system activity. This method has been used alongside resting heart rate and heart rate variability to evaluate ANS response [27,28,29] in various contexts, including lifestyle interventions involving diet and weight loss [11]. Additionally, pupillometry has been applied to assess dietary impact on ANS responses [14] and its association with airway hyperresponsiveness has also been explored [29]. Given its practicality, pupillometry provides a unique opportunity to link dietary-induced ANS changes to physiological outcomes like lung function and airway inflammation.

## 2. Materials and Methods

### 2.1. Participants and Study Design

In this randomized, double-blind, crossover clinical trial, individuals aged 18 to 35 years with or without an asthma diagnosis were eligible to participate if they were classified as either overweight/obese (BMI ≥ 25 kg/m^2^) or had a normal BMI (18.5–24.9 kg/m^2^). The overweight/obese category included both overweight (BMI 25–29.9 kg/m^2^) and obese (BMI ≥ 30 kg/m^2^) individuals. Participants were recruited through trial posters on bulletin boards, newspapers, and internet advertisements and during hospital visits, as previously described [30]. Participants were excluded if they met any of the following criteria: having a respiratory disease other than asthma (except for severe asthma according to GINA guidelines), major systemic diseases (including diabetes, cardiac arrhythmia, angina, congestive heart failure, abnormal electrocardiogram, renal or hepatic failure, malabsorption disease, intestinal inflammatory disease, chronic infectious diseases), or if they were women who were breastfeeding, pregnant, or intending to become pregnant. Additional exclusion criteria included inability to comply with study and follow-up procedures, dietary restrictions (e.g., food allergies, vegetarianism), or being on a weight-loss diet within the three months preceding the study. Seven dropouts occurred during the trial: five due to unavailability to attend the visits before any interventions, two before the second meal—one due to adverse events unrelated to the intervention, and the other due to an inability to attend the study visits. 

Ultimately, 46 individuals were enrolled and randomly assigned to two different isoenergetic but micronutrient-diverse meals, separated by a 7-day washout period. Diet and physical activity were monitored as previously described during the washout period [30]. No significant differences in diet and physical activity were observed between intervention orders (Appendix A).

#### Randomization, Allocation and Sample Size

Participants were randomly assigned to the intervention order in a double-blind fashion, stratified by asthma diagnosis. The sample size was not specifically calculated to assess the autonomic nervous system (ANS) or lung function response to a meal challenge, as these were secondary outcomes of the study protocol [30]. Power calculations were based on the constriction velocity data from a trial evaluating the impact of dietary and exercise interventions on the autonomic nervous system, as no prior studies on the effects of a single meal intervention were available [31]. For lung function and airway inflammation, exhaled nitric oxide (eNO) and FEV1% predicted were considered based on data from a previous trial involving a healthy population challenged with a high-fat meal [32]. For pupillometry, assuming a minimal detectable difference of 0.5 and a within-participant standard deviation of 0.77, a sample size of 40 participants provided 93% power to detect treatment differences using a paired *t*-test at a two-sided significance level of 0.05. For eNO, assuming a minimal detectable difference of −1 and a standard deviation of 4.68, the power was 26%. For FEV1% predicted, with a minimal detectable difference of 0.4 and a standard deviation of 1.1, the power was 61%.

### 2.2. Intervention Protocol

All participants underwent a baseline visit and two interventions, separated by a 7-day washout period. Participants were asked to maintain their usual dietary and physical activity levels during the follow-up and study intervention. On the days of both interventions, participants were requested to have the same breakfast simultaneously to minimize residual effects from breakfast, avoiding coffee or caffeine-containing products. Smokers were asked to abstain from smoking for 12 h before the meal. Additionally, participants were asked to abstain from alcohol consumption during the 7 days preceding the intervention. After each meal, participants were not allowed to eat or drink and were instructed to refrain from physical activity. No significant differences were observed between allocation order in those participating in the protocol [30].

This study was approved on 25 March 2013 by the Ethics Committee of the Faculty of Medicine at Porto University (approval code-CES 318-12), and participants provided written informed consent.

#### Fast Food Meal and Mediterranean Meal

The fast food meal (FFM) was obtained from a typical restaurant, and consisted of a burger with bread, fries, and a sugar-sweetened drink. Nutritional data provided by the food manufacturer indicated that the meal contained total energy (TE): 1026 kcal, carbohydrates: 126 g, protein: 31 g, and total fat: 43 g (12% SFA) [30]. 

The Mediterranean meal (MdM) was prepared in an experimental kitchen using a standardized recipe, available in a previous publication [30]. All ingredients were weighed and preparation methods were always the same and prepared by an independent investigator. The meal was isoenergetic and aligned with the Dietary Reference Intakes (DRI) for macronutrient distribution: carbohydrates 45–65% (added sugars < 25%), protein 10–35%, and fat 20–35% of TE [polyunsaturated fatty acids (PUFA): 5–10%, monounsaturated fatty acids (MUFA): 0.6–1.2%, cholesterol and SFA “as low as possible”) [33]. The MdM included critical features of the Mediterranean diet, specifically a spinach and chickpea soup, whole-wheat pasta and bread, fresh tomatoes with a green salad (arugula and lettuce), olive oil, herbs, sardines, fruit (apple, orange, and dried figs), walnuts, and 40 cL of water as a beverage. Nutritionally, this meal had a TE of 1019 kcal, with 40 g of fat (10.3% SFA), 132 g of carbohydrates, and 37 g of protein. The nutritional data were estimated using the Portuguese Food Composition Table version 4.0 (PortFIR, National Health Institute Doutor Ricardo Jorge, Lisbon, Portugal).

The nutritional characterization of FFM and MdM has been described previously in detail [30].

### 2.3. Outcome Assessment

Outcome measurements were obtained immediately before and 3 h after each meal intake, as previous studies have shown that autonomic nervous system responses and postprandial inflammatory changes typically peak within this time window [32,34]. These measurements included airway inflammation assessed via fractional exhaled nitric oxide (FeNO), lung function assessed through spirometry, and autonomic nervous system function evaluated by pupillometry.

#### 2.3.1. Airway Inflammation

Airway inflammation was assessed by measuring the FeNO using an electrochemical sensor, namely the NObreath^®^ analyzer (Bedfont Scientific, Kent, UK). The procedure involved participants exhaling at a rate of 50 mL/s for 10 s, ensuring a steady flow for at least the last 6 s of the maneuver. The average of three separate measurements was taken to determine the final concentration of exhaled nitric oxide, expressed in parts per billion (ppb), by the American Thoracic Society/European Respiratory Society (ATS/ERS) guidelines [35,36,37].

#### 2.3.2. Lung Function

Lung function was assessed through spirometry using a portable Spirolab spirometer (MIR^®^, Rome, Italy; WINSPIROPRO^®^ software 4.4 version) adhering to ATS data collection standards [38]. Lung function values included forced expiratory volume in the first second (FEV1), forced vital capacity (FVC), forced expiratory flow between 25% and 75% of FVC (FEF25–75%), and peak expiratory flow (PEF). Measurements were considered valid if they met or exceeded the ATS acceptability and reproducibility criteria, specifically: having three acceptable curves present with two reproducible curves and two observed values within 100 mL, or having three acceptable curves present and two reproducible curves with two observed values within 150 mL [39]. 

#### 2.3.3. Autonomic Nervous System

To evaluate the effect of the meals on the ANS, pupillometry was carried out with a portable infrared PLR-200 pupillometer (NeurOptics PLR-200™ Pupillometer, NeurOptics Inc., Irvine, CA, USA) before and 3 h after each meal [40]. Participants had to spend at least 15 min in a semi-dark and quiet room to allow their pupils to adjust to the low light level. After this period, instructions were given to focus on a small object located 3 m away from the eye that was not being measured while keeping their head straight and both eyes open during the measurement. The pupillometer used a light-emitting diode with a single light stimulus at a peak wavelength of 180 nm. A pupillary light response for each eye was recorded, and the average measurement from both the left and right eye was used for statistical analyses. At the end of the measurement cycle, pupil light response curves were recorded, and seven pupillometry outcomes were assessed. These included the initial diameter of the pupil (baseline pupil diameter) and the diameter at the constriction peak (final pupil diameter) in millimeters (mm), relative constriction amplitude (%), average constriction velocity (ACV, mm/s), average dilation velocity (ADV, mm/s), maximum constriction velocity (MCV, mm/s), and the total time taken by the pupil to recover to 75% of its initial resting size after reaching the peak of constriction (T75, seconds) [30]. ADV and T75 are associated with sympathetic nervous system activity, whereas the remaining measurements are related to parasympathetic nervous system activity [30].

### 2.4. Other Procedures

During the baseline assessment, participants completed a sociodemographic and lifestyle habits questionnaire, which included smoking. Clinical asthma diagnoses were determined at baseline according to the Global Initiative for Asthma recommendations [41]. 

Weight and height were evaluated with participants who were lightly clothed and barefoot. Height (cm) was measured using a portable stadiometer (SECA^®^ model 214). Weight (kg) and body composition were measured using a digital scale (Tanita^®^ BC-418 Segmental Body Analyzer, TANITA, Tokyo, Japan). Body mass index (BMI) was calculated as body weight divided by the square of height. Participants were classified according to World Health Organization (WHO) recommendations: underweight (<18.5 kg/m^2^), normal weight (18.5–24.9 kg/m^2^), overweight (25.0–29.9 kg/m^2^), and obese (≥30 kg/m^2^) [42]. 

### 2.5. Statistical Analyses

The characteristics of the participants were presented for the whole sample as percentages for categorical and as median (25th–75th percentile) for continuous variables, except for age and height, which were expressed as mean ± standard deviations (SD). *p*-values < 0.05 were considered statistically significant. Continuous results were expressed as mean (95% confidence interval, CI) or, if not normally distributed, as median 25th–75th. A paired *t*-test was used to assess the dependent measures of individuals with normal distributions. Wilcoxon Signed-Rank Test compared differences in any variables with nonnormal distributions. The relation between groups was performed using the Wilcoxon Signed-Rank Test and paired *t*-test, expressed in the *p*-value. 

We conducted a mixed-effects model analysis to evaluate the effects of meal type (FFM vs. MdM) and time of measurement (After vs. Before meal) on ANS parameters, lung function, and FeNO, accounting for individual variability through random intercepts. Significance levels were set at *p*-value < 0.05. Potential confounders were added, including sex, age, BMI, asthma diagnosis, and smoking status. Interaction effects for variables such as BMI and smoking status were explored but found non-significant; therefore, they were not included in the final model.

All analyses were performed using R software (version 4.0.2, R Foundation for Statistical Computing, Vienna, Austria) and graphs were performed with GraphPad Prism, version 10.0, GraphPad Software, San Diego, CA, USA. 

## 3. Results

The baseline characteristics of the study participants are included in Table 1.

Pupillometry changes after each meal and meal comparisons are summarized in Table 2 and Figure 1. After the MdM intake, there was a significant increase in maximal pupil diameter (MaxD), with a mean variation of 0.17 mm (SD ± 0.50). The average and maximum velocity of constriction (measured by ACV and MCV) were also significantly quicker (absolute variation of velocity 0.12 mm/s [−0.12; 0.32] and 0.25 [−0.07; 0.70], respectively). The MdM was additionally associated with a significant decrease in the median ADV by 0.07 mm/s. Following FFM, there was a decrease in T75 by 1.0 (±1.33) s. 

When comparing the two meals (Figure 1), in the parasympathetic parameters, the MdM led to a more significant increase in both MaxD and MinD than the FFM. Furthermore, SNS parameters differed significantly between meals, with T75 being markedly lower after the FFM.

The results of lung function and FeNO before and after each meal and comparisons between meals are summarized in Table 3. After the MdM intake, there was a reduction in FVC (median variation of −0.06 [−0.17; 0.04]). Following the FFM, there were significant differences in FVC: before, 4.13 (3.45; 5.52) vs. after, 4.19 (3.52; 5.44). No significant change in FEV1 or the FEF 25/75 parameter was seen. Regarding airway inflammation, FeNO increased significantly after MdM by 3.7 [0.4; 9.0] ppb and after FFM 2.2 [−0.3; 9.0] ppb.

When comparing the two meals, there was no significant difference between the MdM and FFM [30]. 

The results of the mixed-effects model analysis for pupillometry are presented in Table 4. The fully adjusted model for sex, age, BMI, asthma diagnosis, and smoke status of MaxD, MinD, %Con, ACV, MCV, ADV, T75, and RHR showed significant baseline effect with an intercept estimated at 8.83 (SE, 0.83), 6.03 (SE, 0.71), −30.36 (SE, 5.03), −5.03 (SE, 0.58), −6.34 (SE, 1.03), 1.08 (SE, 0.23), 3.46 (SE, 0.63), and 79.15 (SE, 12.55), respectively. Both time (before/after meal) and FFM (model 2, estimate = −0.68, *p* ≤ 0.001 and estimate = −0.52, *p* = 0.002, respectively) significantly decreased T75.

Mixed-effects model analyses for lung function and airway inflammation are presented in Table 5. Fully adjusted FEV1/FVC and FEF25/75 models also revealed significant baseline effects with an intercept estimated at 123.91 (SE, 7.59) and 4.61 (SE, 1.35), respectively. 

After adjusting for sex, age, BMI, asthma diagnosis, and smoking status, having a meal, irrespective of the type of meal, had a significant and negative impact on FVC (model 2, estimate = −0.079, *p* < 0.001), FVC predicted, % (model 2, estimate = −1.91, *p* = 0.019) and FEV1 (model 2, estimate = −0.04, *p* = 0.017). FeNO did not show a significant difference, but a tendency of a potential effect of the type of meal was seen (model 2, estimate = −4.21, *p* = 0.057). 

## 4. Discussion

In this randomized crossover clinical trial, we found that different meals elicit distinct autonomic nervous system and airway inflammation responses, even when isoenergetic. A fast food-like meal promoted sympathetic dominance, reflected by a decrease in T75, even after adjusting for relevant confounders. In contrast, a Mediterranean-like meal supported parasympathetic activity, demonstrated by an increase in MaxD and a reduction in ACV and MCV, indicative of a more regulated autonomic state. Both meal types induced a mild reduction in FEV1 and FVC, even after adjusting for confounders. Exhaled nitric oxide tended to differ between meals, suggesting a potential influence on airway inflammation. However, we could not establish an association between airway inflammation and the autonomic nervous system responses.

Eating a meal is associated with increased sympathetic activity [43], likely related to the increased heart rate and cardiac output required for digestion. The specific composition of the meal modulates this response. A high-fat meal impacts the autonomic nervous system and vascular responsiveness [38]; and carbohydrate content can also influence autonomic activity [44]. In our study, although the meals were isoenergetic, they differed in fiber and SFA content. The effects of fiber and fat type on the autonomic nervous system response are unclear, although evidence suggests a potential impact on inflammation. Previous studies have shown that fiber-rich meals, particularly with probiotics, reduce exhaled nitric oxide (FeNO) [45], a marker of airway inflammation, while high-fat meals are linked to increased FeNO [46]. Diets high in SFA and low in antioxidants exacerbate inflammation [2,47,48], reducing lung function and increasing airway inflammation [49]. Additionally, trans-fatty acids and ultra-processed meats have been associated with decreased lung function due to their role in promoting systemic inflammation [50]. It is not possible to say that the observed modulation of the autonomic nervous system and the tendency for increased airway inflammation by a meal are explained by the exact mechanisms underlying how dietary habits and lifestyles impact ANS regulation, lung function, and airway inflammation, as long-term effects have different impacts on health-related outcomes. 

Lifestyle factors like weight loss and dietary composition are key to modulating autonomic control [11,51]. Studies on low-energy diets with varying levels of fiber, red meat, and coffee on autonomic function have not shown significant differences in heart rate variability, possibly due to the small sample sizes and the overshadowing effects of weight loss [52]. Regarding the acute effect of a meal, a small trial showed that high-energy meal intake before dinner promoted sympathetic and decreased parasympathetic activity during sleep [53]. The interaction between diet and the autonomic nervous system (ANS) involves multiple biological pathways, including the gut–brain axis [8]. The vagus nerve plays a central role in mediating bidirectional communication between the gut microbiota and the brain [7]. Through the cholinergic anti-inflammatory pathway, the vagus nerve regulates systemic inflammation by suppressing pro-inflammatory cytokines, such as TNF-α [7]. Diet has a significant influence on vagal activity. For example, high-fat and high-carbohydrate diets have been shown to impair vagus nerve function in mouse models [54]. Additionally, inflammatory mediators such as IL-6, produced by enteric neurons, regulate the number and phenotype of microbe-responsive regulatory T cells in the gut, further linking diet, inflammation, and ANS activity [55]. The nutrient content of a diet, including vitamins, proteins, PUFA, bioflavonoids, carotenoids, and other antioxidant metabolites, contributes to a long-term anti-inflammatory effect [56]. Polyphenols modulate oxidative stress pathways by scavenging reactive oxygen species (ROS) and nitrogen species, thereby reducing LDL oxidation [57]. They also promote anti-inflammatory pathways by inactivating nuclear factor kappa B (NF-κB), downregulating pro-inflammatory cytokines such as TNF-α and IL-6, and reducing the release of arachidonic acid, prostaglandins, and leukotrienes via the arachidonic acid signaling pathway [58]. Similarly, omega-3 polyunsaturated fatty acids (*n*-3 PUFA) exhibit anti-inflammatory effects by reducing levels of CRP, IL-6, and TNF-α, primarily through the inhibition of NF-κB activation and oxidative stress [59]. Furthermore, previous studies have shown that patients with low plasma levels of *n*-3 PUFA display an unbalanced pro-sympathetic response [60]. As oxidative stress promotes sympathoexcitatory effects, the use of antioxidants and anti-inflammatory agents, such as polyphenols and *n*-3 PUFA, may help mitigate adrenergic overdrive and restore autonomic balance [57]. A typical MdM, rich in these compounds, contributed to vagal activation, promoting a parasympathetic state, as evidenced by decreases in ACV and MCV [61]. Nevertheless, few studies have evaluated the effects of a meal. A trial comparing the acute effect of high-fat and high-carbohydrate isoenergetic meals promoted differential effects on autonomic response in lean versus obese women [62]. Acute effects may depend on factors such as fitness status, dietary habits, and dietary diversity. Our previous work suggested that a more widely varied vegetable intake was linked to sympathetic activity [14], possibly due to the synergistic effects of dietary components with anti-inflammatory and antioxidant properties. Conversely, Western diets, typically high in SFA and deficient in essential minerals and vitamins, including vitamin B12 and vitamin A, have been associated with autonomic dysfunction (AD) and inflammation [63,64]. Although the exact mechanisms by which oxidative stress influences sympathoexcitatory effects remain unclear [57], our study provides evidence that even a single meal might immediately impact autonomic balance. Previous studies have suggested that the components of these diets, including dietary fiber, micronutrients, and bioactive compounds, play critical roles in gut microbiota activity, reducing low-grade inflammation and protecting cells from oxidative stress [65]. Although the effect of meals on the microbiome has been contradictory, it has been shown that a meal and the timing of a meal might also influence the microbiota population [66]. Together, these processes may influence both inflammation and ANS function. 

A diet rich in anti-inflammatory and antioxidant nutrients, such as those found in Mediterranean food, has been shown to positively influence lung function by reducing airway inflammation and oxidative stress [67,68]. Key dietary components, such as polyphenols and omega-3 fatty acids, may protect lung function [47,69]. Our mixed-model analysis showed no significant differences between the MdM and FFM regarding FeNO levels. The absence of significant differences may be partially explained by the limited sample size, which reduced the power to detect small effect sizes. Mechanistically, postprandial inflammatory responses could play a role. High-fat meals have been associated with postprandial systemic inflammation, marked by increased levels of IL-6, TNF-α, and triglyceride [24,70], which have been correlated with increased airway FeNO levels [70]. Neutrophilic airway inflammation patterns have been linked to high-fat meals, which also suppress bronchodilator recovery in individuals with asthma [32]. Additionally, the gut–lung axis may influence FeNO regulation, as dietary impacts on the gut microbiota can affect systemic inflammation [48] However, the acute nature of this study may have limited the potential for significant microbiota-mediated effects. Greater vegetable diversity has been associated with lower airway inflammation [18], likely due to the antioxidant properties of these foods [48]. Dietary patterns rich in animal proteins and carbohydrates have been linked to increased asthma symptoms and reduced lung function [68]. While the two tested meals differed in micronutrient composition, their comparable fat content could have minimized differences in postprandial airway inflammation and in lung function. Previous studies have evaluated specific interventions in meal-induced airway inflammation, such as avoiding exercise-induced airway inflammation [8] or supplementing meals with fibers and probiotics [9], which may have enhanced their ability to detect differences. A meal is likely to lead to acute changes dependent on concomitant factors, stress, exercise, and individual fitness. A previous study demonstrated that consuming a meal classified as ‘anti-inflammatory’ according to the Dietary Inflammatory Index^®^ significantly enhanced the exercise-induced reduction in airway inflammation, particularly in reducing sputum eosinophils, in individuals with asthma [71]. Diets rich in fruits and vegetables, fiber, healthy fats, and other high quality dietary patterns appear to benefit individuals with asthma, specially obese individuals [48]. While the acute nature of this study limits conclusions about long-term effects, these findings underscore the importance of also assessing acute dietary interventions and their impact on asthma outcomes.

It is essential to interpret our results with caution due to several limitations. This study focused on young, healthy adults; although we included some participants with asthma and obesity, their numbers were insufficient for subgroup analysis. Nevertheless, we adjusted our models for these co-founders and found a significant difference between interventions. The sample size was insufficiently powered to assess airway inflammation and lung function. However, we observed a tendency for both parameters to change in response to a meal. The findings of our study should not be generalized to broader claims about the Mediterranean diet. Our meals differed primarily in micronutrient composition, which aligns with typical dietary habits but may yield different results than meals with more significant macronutrient variations, especially in fat and antioxidants content. Nonetheless, as the FFM was sourced from a commercial restaurant, its nutritional composition was derived from the manufacturer’s database. It is possible that the parasympathetic activity seen after the MdM, characterized by increased MaxD, ACV, and MCV, might be diluted by the low-impact bioactive compounds and antioxidant profile of the FFM in modulating autonomic responses. Heart rate variability (HRV), an established marker of ANS function [11], could provide complementary insights into our pupillometry results. HRV reflects not only the balance of ANS response but also the impact of food on ANS parameters, particularly for short-term HRV measurements [72]. It has been closely associated with bronchial hyperresponsiveness [29]. However, the combined use of HRV and pupillometry to assess the effects of diet or meal interventions remains unexplored, and further standardization of both methods is needed [72]. We observed minor differences in sex and smoking status across participants; adjustments were made to account for these factors, and even with these adjustments, meal type still significantly influenced autonomic responses. The demographic homogeneity of our study population, with 94% of participants identifying as Caucasian, limits the generalizability of our findings to more diverse populations. Previous studies have reported race-based differences in autonomic cardiovascular regulation in African Americans, which may influence autonomic responses to interventions [73]. Future research should aim to include more diverse populations to validate these findings and explore potential racial or ethnic differences in autonomic nervous system regulation. During the washout, participants were asked to maintain their regular dietary and physical activity habits, reducing potential confounding factors from lifestyle changes. No further significant differences were found between intervention orders (Appendix A). Although 57% of our participants were women, we did not monitor or control for menstrual cycle phase, which may influence autonomic nervous system responses. A previous study suggested that certain phases of the menstrual cycle might be associated with lower parasympathetic and higher sympathetic activity following a high-energy meal. This factor could potentially modulate the autonomic response to dietary interventions and should be monitored [53]. Future studies should also assess how long the observed effects persist and how subsequent meals impact ANS function and lung function. Long-term adherence to a nutritionally rich plant-centered diet has been associated with slower lung function decline in a longitudinal cohort study [3]. These diets may exert cumulative benefits through sustained anti-inflammatory and antioxidant effects, which can influence systemic inflammation and oxidative stress over time. Additionally, a previous systematic review highlighted that the duration of the dietary interventions influenced weight change and autonomic nervous system function [11], suggesting that repeated dietary exposure might promote a progressive modulation of the autonomic nervous system. The short intervention with a washout period reduced potential carryover effects. In addition, although important confounders have been considered, we cannot exclude the possible impact of other unmeasured covariates (i.e., environment) [13]. 

This study’s strengths include the rigorous randomized crossover design, which minimized confounding factors and balanced baseline differences. The meals were designed to be isoenergetic and aligned with dietary reference intakes for macronutrient distribution, which allowed us to compare the micronutrient content effectively. Pupillometry, which measures ANS responses under controlled lighting conditions, provides reliable data. Similar studies have demonstrated that pupillometry outcomes align with HRV measurements [74] and have been used successfully in dietary intervention research [63].

## 5. Conclusions

In conclusion, our study showed that a single meal, depending on its quality, can significantly impact autonomic nervous system responses: the Mediterranean meal had a protective effect, inducing a parasympathetic response in comparison to a fast food meal’s sympathetic response. Although airway inflammation and lung function changes were not clinically significant, it is likely that in specific populations, namely obese and asthmatic, this study might render different results. The predominance of sympathetic activity following meals, which we also observed here after an FFM, could play a role in the development of hypertension, cardiovascular disease, and other obesity-related complications, highlighting the need for interventions that focus on both dietary quality and ANS regulation. The study findings highlight the role of meal quality in autonomic regulation, with potential implications for cardiometabolic and respiratory health. This study highlights the importance of daily dietary choices, demonstrating that meal quality acutely influences autonomic nervous system responses and airway inflammation. From a societal perspective, these results emphasize the need for policies that promote easy access to healthier dietary options, particularly in environments where fast-food consumption is prevalent. Clinically, the study suggests that dietary interventions emphasizing balanced meals rich in unsaturated fats and antioxidants could support parasympathetic activation and mitigate sympathetic overdrive, which is associated with obesity-related complications and respiratory conditions such as asthma [48]. Further research is needed to explore the cumulative effects of the subsequent meals, increasing awareness of the health impact of our daily dietary choices.

## Figures and Tables

**Figure 1 nutrients-17-00614-f001:**
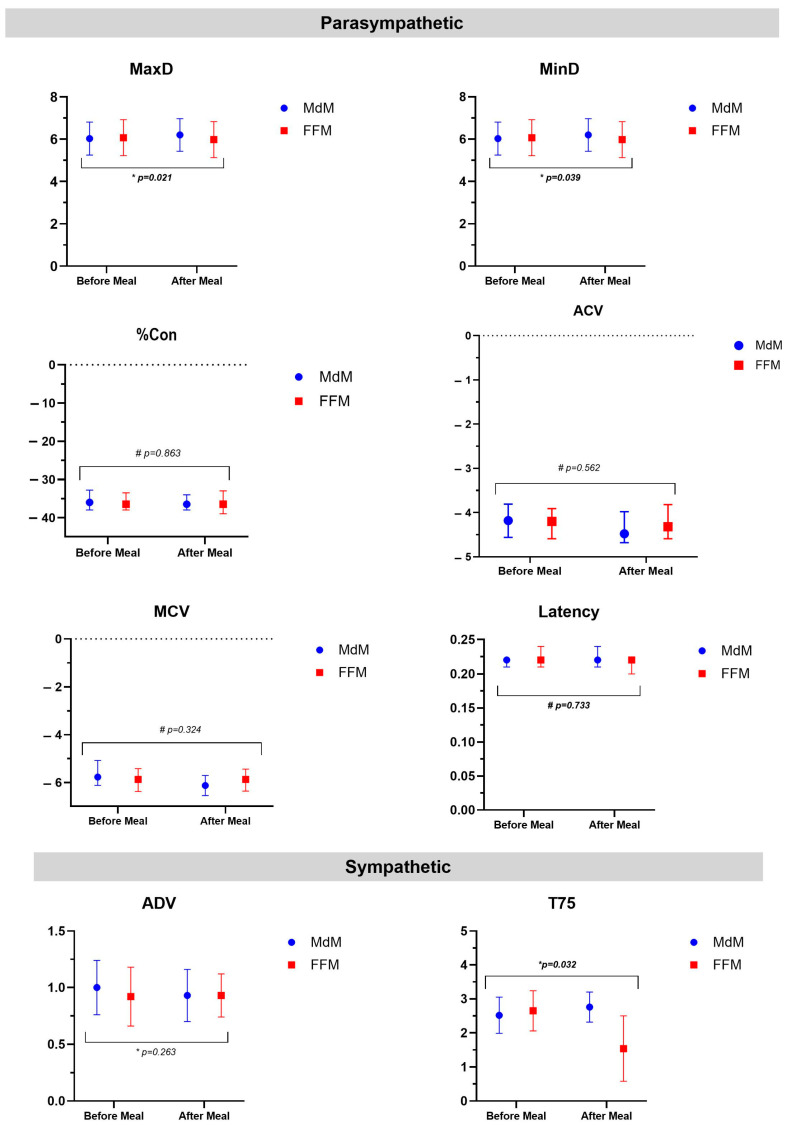
Pupillometry parameter changes before and after each meal and comparison between meals. MdM—Mediterranean Meal; FFM—Fast Food Meal. MaxD, maximal diameter; MinD, minimum diameter; ADV, dilation velocity, T75, time at which pupil has re-dilated 75% of the reflex amplitude are presented as mean and standard deviation; for %Con, percent of the constriction; Latency, time of the onset of the constriction; ACV, average constriction velocities; MCV, maximum constriction velocity data is presented as median and interquartile ratio. *: Paired samples *t*-test; ^#^: Wilcoxon Signed-Rank Test.

**Table 1 nutrients-17-00614-t001:** Participants’ characteristics (*n* = 46).

	Total
Female	26 (57)
Age, years (median, 25th–75th)	25, 22–30
Caucasian	43 (94)
Height, cm (mean ± SD)	169 ± 10.4
BMI	
Average weight (18.5–24.9 kg/m^2^)	25 (54)
Overweight (25–29.9 kg/m^2^)	17 (37)
Obese (≥30 kg/m^2^)	4 (9)
Active smokers	7 (15)
Asthma diagnosis	13 (28.3)

Data are presented as *n* (%), except if otherwise specified. Based on a previous publication [30].

**Table 2 nutrients-17-00614-t002:** Pupillometry parameter changes before and after each meal.

	MdM			FFM			∆ Between Meals
Measurements	Before	After	*p*	Before	After	*p*	
**Parasympathetic**							
MaxD, mm (mean ± SD)	6.03 ± 0.78	6.20 ± 0.77	0.043 ^a^	6.07 ± 0.85	5.98 ± 0.85	0.249 ^a^	0.26 ± 0.71
MinD, mm (mean ± SD)	3.93 ± 0.62	3.97 ± 0.60	0.128 ^a^	3.92 ± 0.66	3.87 ± 0.72	0.352 ^a^	0.17 ± 0.51
%Con (%)	−36.0 (−38.0; −32.8)	−36.5 (−38.0; −34.0)	0.072 ^b^	−36.5 (−38.0; −33.5)	−36.5 (−39.0; −33.0)	0.467 ^b^	0.50 (−2.00; 2.50)
Latency, s	0.22 (0.21; 0.22)	0.22 (0.21; 0.24)	0.461 ^b^	0.22 (0.21; 0.24)	0.22 (0.20; 0.22)	0.876 ^b^	0.0 (−0.015; 0.015)
ACV, mm/s	−4.18 (−4.56; −3.81)	−4.48 (−4.68; −3.98)	0.035 ^b^	−4.20 (−4.59; −3.91)	−4.32 (−4.59; −3.82)	0.483 ^b^	−0.060 (−0.29; 0.26)
MCV, mm/s	−5.77 (−6.12; −5.08)	−6.13 (−6.55; −5.71)	0.003 ^b^	−5.87 (−6.38; −5.42)	−5.87 (−6.36; −5.44)	0.324 ^b^	−0.03 (−0.85; 0.37)
**Sympathetic**							
ADV, mm/s (mean ± SD)	1.00 ± 0.24	0.93 ± 0.23	0.030 ^a^	0.92 ± 0.26	0.93 ± 0.19	0.412 ^a^	1.62 ± 9.53
T75, s (mean ± SD)	2.52 ± 0.53	2.76 ± 0.44	0.454 ^a^	2.65 ± 0.59	1.54 ± 0.96	0.012 ^a^	1.84 ± 0.67

Data are presented as a median and 25th–75th, except if specified otherwise. ^a^: Paired samples *t*-test; ^b^: Wilcoxon Signed-Rank Test. MaxD, maximal diameter; MinD, minimum diameter; %Con, percent of the constriction; latency, time of the onset of the constriction; ACV, average constriction velocities; MCV, maximum constriction velocity; ADV, dilation velocity; T75, time at which pupil has re-dilated 75% of the reflex amplitude.

**Table 3 nutrients-17-00614-t003:** Lung function and airway inflammation changes before and after each meal, and comparison between meals.

	MdM			FFM		
Measurements	Before	After	*p*	Before	After	*p*
**Lung function**						
FVC, median (25th; 75th)	4.08 (3.55; 5.40)	4.08 (3.44; 4.98)	0.023 ^b^	4.13 (3.45; 5.52)	4.19 (3.52; 5.44)	0.029 ^b^
FEV1, median (25th; 75th)	3.67 (2.93; 4.44)	3.58 (2.94; 4.41)	0.055 ^b^	3.65 (3.03; 4.48)	3.73 (3.00; 4.45)	0.245 ^b^
FEV1/FVC	85.7 ± 5.43	86.2 ± 5.20	0.311 ^a^	85.6 ± 5.70	86.3 ± 5.29	0.118 ^a^
FEF25/75	4.0 ± 1.11	4.09 ± 1.21	0.110 ^a^	4.11 ± 1.10	4.09 ± 1.13	0.817 ^a^
**Airway inflammation**						
FeNO, ppb, median (25th; 75th)	25.7 (16.3; 49.3)	30.8 (25.6; 62.3)	<0.001 ^b^	25.0 (14.3; 47.0)	29.3 (17.0; 51.7)	0.040 ^b^

Data are presented as mean ± SD, except if specified otherwise. ^a^: Paired samples *t*-test; ^b^: Related-Wilcoxon signed rank test. *p*: between the time of measure. FVC forced vital capacity; FEV1 forced expiratory volume in 1 s; FEF25–75 forced expiratory flow middle portion of forced vital capacity; FeNO, the fraction of exhaled nitric oxide.

**Table 4 nutrients-17-00614-t004:** Mixed-effects model analysis for pupillometry and resting heart rate.

		Fixed Effects			Random Effects		Model Fit		
		Intercept	Meal FFM vs. MdM	Time After vs. Before	Intercept	Residual	Log-Likelihood	AIC	BIC
**MaxD**									
	Model 1								
	Estimate	6.09	−0.08	0.04	--	--	−138.56	287.13	302.50
	SE	0.12	0.07	0.06	--	--	--	--	--
	Variance	--	--	--	0.47	0.17	--	--	--
	SD	--	--	--	0.68	0.41	--	--	--
	DF	53.81	117.99	117.19	--	--	--	--	--
	* p*	<0.001	0.215	0.508	--	--	--	--	--
	Model 2								
	Estimate	8.83	−0.08	0.04	--	--	−140.15	300.30	331.05
	SE	0.83	0.07	0.06	--	--	--	--	--
	Variance	--	--	--	0.40	0.17	--	--	--
	SD	--	--	--	0.63	0.41	--	--	--
	DF	35.27	117.89	117.22	--	--	--	--	--
	* p*	<0.001	0.222	0.508	--	--	--	--	--
**MinD**									
	Model 1								
	Estimate	3.96	−0.06	0.01	--	--	−98.90	207.79	223.04
	SE	0.09	0.05	0.05	--	--	--	--	--
	Variance	--	--	--	0.33	0.10	--	--	--
	SD	--	--	--	0.58	0.32	--	--	--
	DF	51.20	113.80	113.05	--	--	--	--	--
	* p*	<0.001	0.231	0.792	--	--	--	--	--
	Model 2								
	Estimate	6.03	−0.06	0.01	--	--	−102.11	224.22	254.72
	SE	0.71	0.05	0.05	--	--	--	--	--
	Variance	--	--	--	0.30	0.10	--	--	--
	SD	--	--	--	0.55	0.32	--	--	--
	DF	35.03	113.76	113.19	--	--	--	--	--
	* p*	<0.001	0.233	0.788	--	--	--	--	--
**%Con**									
	Model 1								
	Estimate	−35.20	−0.13	−0.21	--	--	−376.14	762.28	777.60
	SE	0.64	0.29	0.29	--	--	--	--	--
	Variance	--	--	--	14.53	3.81	--	--	--
	SD	--	--	--	3.29	1.82	--	--	--
	DF	48.58	115.53	114.96	--	--	--	--	--
	* p*	<0.001	0.651	0.470	--	--	--	--	--
	Model 2								
	Estimate	−30.36	−0.13	−0.21	--	--	−372.72	765.43	796.06
	SE	5.03	0.29	0.29	--	--	--	--	--
	Variance	--	--	--	15.39	3.29	--	--	--
	SD	--	--	--	3.92	1.82	--	--	--
	DF	34.83	115.30	114.91	--	--	--	--	--
	* p*	<0.001	0.652	0.474	--	--	--	--	--
**ACV**									
	Model 1								
	Estimate	−4.21	0.13	−0.04	--	--	−144.81	299.62	314.62
	SE	0.09	0.08	0.08	--	--	--	--	--
	Variance	--	--	--	0.17	0.25	--	--	--
	SD	--	--	--	0.41	0.50	--	--	--
	DF	86.37	118.36	116.42	--	--	--	--	--
	* p*	<0.001	0.109	0.629	--	--	--	--	--
	Model 2								
	Estimate	−5.03	0.12	−0.04	--	--	−150.01	320.01	350.01
	SE	0.58	0.08	0.08	--	--	--	--	--
	Variance	--	--	--	0.15	0.25	--	--	--
	SD	--	--	--	0.39	0.50	--	--	--
	DF	34.84	117.40	115.76	--	--	--	--	--
	* p*	<0.001	0.127	0.624	--	--	--	--	--
**MCV**									
	Model 1								
	Estimate	−5.78	−0.003	−0.28	--	--	−245.37	500.73	516.08
	SE	0.16	0.16	0.16	--	--	--	--	--
	Variance	--	--	--	0.35	1.01	--	--	--
	SD	--	--	--	0.59	1.01	--	--	--
	DF	107.93	119.07	116.67	--	--	--	--	--
	* p*	<0.001	0.984	0.079	--	--	--	--	--
	Model 2								
	Estimate	−6.34	−2.61 × 10^−3^	−0.28	--	--	−250.77	521.54	552.23
	SE	1.03	0.16	0.16	--	--	--	--	--
	Variance	--	--	--	0.41	0.64	--	--	--
	SD	--	--	--	1.01	1.01	--	--	--
	DF	35.55	118.34	116.43	--	--	--	--	--
	* p*	<0.001	0.987	0.079	--	--	--	--	--
**ADV**									
	Model 1								
	Estimate	0.97	−0.02	−0.03	--	--	167.48	−22.95	−7.93
	SE	0.03	0.03	0.03	--	--	--	--	--
	Variance	--	--	--	0.03	0.03	--	--	--
	SD	--	--	--	0.16	0.17	--	--	--
	DF	74.79	108.15	107.24	--	--	--	--	--
	* p*	<0.001	0.440	0.229	--	--	--	--	--
	Model 2								
	Estimate	1.08	−0.02	−0.03	--	--	5.35	9.29	39.33
	SE	0.23	0.03	0.03	--	--	--	--	--
	Variance	--	--	--	0.03	0.03	--	--	--
	SD	--	--	--	0.16	0.17	--	--	--
	DF	33.54	107.01	106.35	--	--	--	--	--
	* p*	<0.001	0.482	0.233	--	--	--	--	--
**T75**									
	Model 1								
	Estimate	2.98	−0.67	−0.54	--	--	−97.76	205.52	217.56
	SE	0.17	0.18	0.18	--	--	--	--	--
	Variance	--	--	--	0.16	0.61	--	--	--
	SD	--	--	--	0.17	0.78	--	--	--
	DF	79.0	79.0	79.0	--	--	--	--	--
	* p*	<0.001	<0.001	<0.001	--	--	--	--	--
	Model 2								
	Estimate	3.46	−0.68	−0.52	--	--	−97.63	215.27	239.34
	SE	0.63	0.16	0.17	--	--	--	--	--
	Variance	--	--	--	0.10	0.53	--	--	--
	SD	--	--	--	0.07	0.73	--	--	--
	DF	74.0	74.0	74.0	--	--	--	--	--
	* p*	<0.001	<0.001	0.002	--	--	--	--	--

Data are presented: β-coefficients, *p* significance < 0.05. Non-adjusted Model (Model 1): Fixed effects with intercept: (Meal + Time of measurement) & Random effects with intercept: factor (ID); Adjusted Model (Model 2): Fixed effects with intercept: (Meal + Time of measure + gender + age + BMI + Asthma diagnosis + smoke status) & Random effects with intercept: factor (ID). MaxD, maximal diameter; MinD, minimum diameter; %Con, percent of the constriction; ACV, average constriction velocities; MCV, maximum constriction velocity; ADV, dilation velocity; T75, time at which pupil has re-dilated 75% of the reflex amplitude; RHR, rest heart rate; MdM—Mediterranean Meal; FFM—Fast Food Meal.

**Table 5 nutrients-17-00614-t005:** Mixed-effects model analysis for lung function and airway inflammation.

		Fixed Effects			Random Effects		Model Fit		
		Intercept	Meal FFM vs. MdM	Time After vs. Before	Intercept	Residual	Log-Likelihood	AIC	BIC
**FVC**									
	Model 1								
	Estimate	4.33	0.039	−0.079	--	--	−629.08	1302.15	1369.67
	SE	0.16	0.023	0.023	--	--	--	--	--
	Variance	--	--	--	1.027	0.021	--	--	--
	SD	--	--	--	1.013	0.146	--	--	--
	DF	40.82	117.07	117.01	--	--	--	--	--
	* p*	<0.001	0.097	<0.001	--	--	--	--	--
	Model 2								
	Estimate	1.63	0.039	−0.079	--	--	−14.72	75.44	146.17
	SE	1.35	0.023	0.023	--	--	--	--	--
	Variance	--	--	--	0.963	0.021	--	--	--
	SD	--	--	--	0.981	0.146	--	--	--
	DF	21.98	117.01	116.98	--	--	--	--	--
	* p*	0.238	0.096	<0.001	--	--	--	--	--
**FEV1**									
	Model 1								
	Estimate	3.70	0.02	−0.04	--	--	−75.21	160.42	170.53
	SE	0.13	0.02	0.02	--	--	--	--	--
	Variance	--	--	--	0.66	0.01	--	--	--
	SD	--	--	--	0.81	0.11	--	--	--
	DF	40.76	117.06	117.0	--	--	--	--	--
	* p*	<0.001	0.186	0.017	--	--	--	--	--
	Model 2								
	Estimate	2.18	0.02	−0.04	--	--	−14.72	75.44	146.17
	SE	1.15	0.017	0.018	--	--	--	--	--
	Variance	--	--	--	0.71	0.84	--	--	--
	SD	--	--	--	0.013	0.11	--	--	--
	DF	21.99	117.02	116.99	--	--	--	--	--
	* p*	0.072	0.186	0.017	--	--	--	--	--
**FEV1/FVC**									
	Model 1								
	Estimate	102.52	0.32	0.83	--	--	−430.14	870.28	885.63
	SE	1.0	0.39	0.39	--	--	--	--	--
	Variance	--	--	--	36.38	5.90	--	--	--
	SD	--	--	--	6.03	2.43	--	--	--
	DF	18.76	113.45	114.32	--	--	--	--	--
	* p*	<0.001	0.411	0.033	--	--	--	--	--
	Model 2								
	Estimate	123.91	0.31	−2.31	--	--	−379.58	805.17	875.75
	SE	7.59	0.39	1.21	--	--	--	--	--
	Variance	--	--	--	29.13	5.90	--	--	--
	SD	--	--	--	5.40	2.43	--	--	--
	DF	22.12	116.39	22.0	--	--	--	--	--
	* p*	<0.001	0.433	0.06	--	--	--	--	--
**FEF25/75**									
	Model 1								
	Estimate	4.03	0.005	0.04	--	--	−204.30	418.59	434.38
	SE	0.18	0.04	0.04	--	--	--	--	--
	Variance	--	--	--	1.22	0.07	--	--	--
	SD	--	--	--	1.11	0.26	--	--	--
	DF	42.27	117.17	117.01	--	--	--	--	--
	* p*	<0.001	0.905	0.349	--	--	--	--	--
	Model 2								
	Estimate	4.61	0.004	0.830	--	--	−104.90	219.80	235.18
	SE	1.35	0.04	0.39	--	--	--	--	--
	Variance	--	--	--	1.26	1.12	--	--	--
	SD	--	--	--	0.07	0.26	--	--	--
	DF	22.03	117.13	116.1	--	--	--	--	--
	* p*	0.002	0.922	0.349	--	--	--	--	--
**FeNO**									
	Model 1								
	Estimate	44.08	−4,15	3.79	--	--	−705.05	1422.09	1440.51
	SE	6.81	2.19	2.16	--	--	--	--	--
	Variance	--	--	--	1758.83	185.82	--	--	--
	SD	--	--	--	41.94	13.63	--	--	--
	DF	44.19	116.37	116.08	--	--	--	--	--
	* p*	<0.001	0.058	0.083	--	--	--	--	--
	Model 2								
	Estimate	−7.01	−4.22	3.80	--	--	−629.08	1302.15	1369.67
	SE	56.95	2.19	2.16	--	--	--	--	--
	Variance	--	--	--	1776.84	42.15	--	--	--
	SD	--	--	--	185.82	13.63	--	--	--
	DF	23.08	116.07	116.07	--	--	--	--	--
	* p*	0.903	0.057	0.082	--	--	--	--	--

Data are presented: β-coefficients, *p* significance < 0.05. Non-adjusted Model (Model 1): Fixed effects with intercept: (Meal + Time of measure) & Random effects with intercept: factor (ID) Adjusted Model (Model 2): Fixed effects with intercept: (Meal + Time of measure + gender + age + BMI + Asthma diagnosis + smoke status) & Random effects with intercept: factor (ID). FVC forced vital capacity; FEV1 forced expiratory volume in 1 s; FEF25-75 forced expiratory flow middle portion of forced vital capacity; FeNO, a fraction of exhaled nitric oxide; MdM—Mediterranean Meal; FFM—Fast Food Meal.

## Data Availability

The data presented in this study are available on request from the corresponding author due to the confidential nature of the data used in this study.

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
