# Peer review of "The Acute Effects of a Fast-Food Meal Versus a Mediterranean Food Meal on the Autonomic Nervous System, Lung Function, and Airway Inflammation: A Randomized Crossover Trial"

_nutrients, 2025, doi:10.3390/nu17040614_

Round 1
Reviewer 1 Report
Comments and Suggestions for Authors
This study evaluated the acute effects of two meals, a Mediterranean-like meal and a fast food-like meal, because despite being isoenergetic, they may have different micronutrient compositions. The text was well conducted and presents relevant results thanks to its approach involving the relationship between the nervous system, nutrition, inflammation and lung function, which gives an interdisciplinary characteristic to the study. However, there are points that need to be improved.
1. In the introduction, the mechanistic relationship between diet, the autonomic nervous system (ANS) and lung function could be better detailed.
2. The justification for using pupillometry as the main indicator could be strengthened.
3. The authors can explain how the sample calculation was done, how the number of participants was defined and the implications of the number for the interpretation and impact of the results.
4. The authors should detail the methods of meal preparation. 5. The inclusion of a 7-day washout period is positive, but the lack of rigorous monitoring of participants' diet and physical activity during this period limits internal validity. It would be interesting to hear the authors' opinion on this.
6. The use of pupillometry is interesting, but the validation of this method as a direct indicator of ANS responses in a dietary context is not discussed in depth.
7. Heart rate variability data could have complemented the findings. It would be interesting if the authors included this information and discussion in the text.
8. The absence of significant differences in FeNO between meals partially contradicts the initial hypothesis. Could the authors discuss this result in more detail, including possible mechanisms involved in FeNO regulation. For example, could the production of postprandial inflammatory mediators or changes in the microbiota play a relevant role?
9. The impact of specific micronutrients, such as polyphenols and omega-3 fatty acids, on the ANS is discussed, but the interaction of these compounds with inflammatory pathways could be further explored and better discussed.
10. Demographic homogeneity (94% Caucasian) reduces the generalizability of the results. What can the authors say about this?
11. The regression analysis adjusted for variables such as obesity and smoking is adequate, but an interaction analysis exploring the influence of these factors is missing.
12. The authors could expand the discussion on the biological mechanisms that link diet and ANS, including the role of the gut-brain axis and inflammatory mediators such as TNF-α and IL-6.
13. The authors did some more detailed subgroup analysis, highlighting differences between asthmatics, obese individuals, and non-obese individuals.
14. It would be very interesting if the authors discussed in more detail the possible cumulative effects of repeated meals on ANS and lung function.
15. The authors could include a brief description of the societal impact of the study and the implications of the findings for the population studied.
16. Avoid generalized statements about the benefits of Mediterranean diets. In this sense, the limitations of the study can be improved.
17. It would be interesting to hear from the authors the importance of longitudinal studies on the subject of this study, if any.
Author Response
Response to Reviewer #1:
This study evaluated the acute effects of two meals, a Mediterranean-like meal and a fast food-like meal, because despite being isoenergetic, they may have different micronutrient compositions. The text was well conducted and presents relevant results thanks to its approach involving the relationship between the nervous system, nutrition, inflammation and lung function, which gives an interdisciplinary characteristic to the study. However, there are points that need to be improved.
Comment 1. In the introduction, the mechanistic relationship between diet, the autonomic nervous system (ANS) and lung function could be better detailed.
Response 1: Thank you for your suggestion. We have revised the introduction to include a more detailed proposal of a potential mechanistic relationship between diet, the ANS, and lung function. This statement is based not only on some specific dietary components, like omega-3 and polyphenols, but also on our previous studies on dietary diversity. The revised text is included in the Introduction, page 2 from lines 60-66, and reads as follows: “In school-aged children, dietary diversity has been associated with decreased airway inflammation [18] and modulation of ANS responses [14]. Airway relaxation, mediated by the activation of β-receptors in the sympathetic pathway, may be influenced by dietary components such as omega-3 polyunsaturated fatty acids (PUFA), [19] or polyphenols [20]. By modulating ANS balance, dietary components and diversity might influence lung function. The ANS response to a meal may also vary according to physical activity level [21] and obesity [22].”
Comments 2. The justification for using pupillometry as the main indicator could be strengthened.
Response 2: Thank you for the valuable comment. We have revised the manuscript to strengthen the justification for using pupillometry as a main indicator of ANS responses. Specifically. its impact and previous studies addressing lifestyle interventions and diet, in order to comply to the comment 6. The revised text in the introduction, line 81-89 now reads: “By measuring pupil dynamics, such as constriction and dilation, it reflects the balance between sympathetic and parasympathetic nervous system activity. This method has been used alongside resting heart rate and heart rate variability to evaluate ANS response [27-29] in various contexts, including lifestyle interventions involving diet and weight loss [11]. Additionally, pupillometry has also been applied to assess dietary impact on ANS responses [14], and its association with airway hyperresponsiveness has also been explored [29]. Given its practicality, pupillometry provides a unique opportunity to link dietary-induced ANS changes to physiological outcomes like lung function and airway inflammation.”
Comment 3. The authors can explain how the sample calculation was done, how the number of participants was defined and the implications of the number for the interpretation and impact of the results.
Response 3: Thank you for your insightful comment. We have included a new subchapter in the methods section to clarify the sample size calculation and its implications for the results. The sample size was not specifically calculated for autonomic nervous system (ANS) or lung function outcomes, as these were secondary outcomes of the study protocol. To address this, we calculated the power for detecting differences in each outcome. However, we acknowledge that the study's exploratory nature limits these calculations. The sample size calculation for the autonomic nervous system was based on the constriction velocity outcome, using data from a previous trial assessing the impact of dietary and exercise interventions on the ANS, as no prior studies had evaluated meal interventions. Exhaled nitric oxide (eNO) and FEV1 % predicted were used for airway inflammation and lung function, respectively. The calculations were based on a previous trial involving healthy participants challenged with a high-fat meal.
We have included the following sub-chapter in the methods section (lines 124-139): “2.1.1. Randomization, allocation, and sample size: Participants were randomly assigned to the intervention order in a double-blinded fashion, stratified by asthma diagnosis. The sample size was not specifically calculated to assess the autonomic nervous system (ANS) or lung function response to a meal challenge, as these were secondary outcomes of the study protocol[30]. Power calculations were based on the constriction velocity data from a trial evaluating the impact of dietary and exercise interventions on the autonomic nervous system, as no prior studies on the effects of a single meal intervention were available[31]. For lung function and airway inflammation, exhaled nitric oxide (eNO) and FEV1 % predicted were considered based on data from a previous trial involving a healthy population challenged with a high-fat meal[32]. For pupillometry, assuming a minimal detectable difference of 0.5 and a within-participant standard deviation of 0.77, a sample size of 40 participants provided 93% power to detect treatment differences using a paired t-test at a two-sided significance level of 0.05. For eNO, assuming a minimal detectable difference of -1 and a standard deviation of 4.68, the power was 26%. For FEV1 % predicted, with a minimal detectable difference of 0.4 and a standard deviation of 1.1, the power was 61%.”
In the discussion section, we reinforced the impact of sample size on our results (lines 463-464): “The sample size was insufficiently powered to assess airway inflammation and lung function. However, we observed a tendency for both parameters to change in response to a meal.”
Comment 4. The authors should detail the methods of meal preparation.
Response 4: Thank you for this suggestion. Details regarding meal preparation, including the recipe, have been previously described and published in the research protocol (Silva, D., et al., Meal-exercise challenge and physical activity reduction impact on immunity and inflammation (MERIIT trial), Contemp Clin Trials Commun, 2018. 10: p. 177-189), which is available as an open-access publication. We believe this reference to the published protocol sufficiently details the meal preparation methods while maintaining conciseness in the manuscript. Nevertheless, we have now specified this in the methods section and included additional information regarding the preparation process (lines 162-163):“The Mediterranean meal (MdM) was prepared in an experimental kitchen using a standardized recipe, available at a previous publication[30], all ingredients were weighted and preparation methods were always the same and prepared by an independent investigator.”
Comment 5. The inclusion of a 7-day washout period is positive, but the lack of rigorous monitoring of participants' diet and physical activity during this period limits internal validity. It would be interesting to hear the authors' opinion on this.
Response 5: Thank you for this comment. We appreciate the opportunity to clarify this aspect of the study design. Besides having a 7-day washout period, we performed dietary and physical activity monitoring during this period using a three-day estimated food record and physical activity monitor using an accelerometer and a pedometer. The methodology used has been previously described in detail [1]. There were no significant differences between intervention order between the two groups. As this is a cross-over study where each participant acted as their own control, inter-individual variability was also inherently controlled, which further reduces concerns regarding the impact on internal validity. To clarify in the manuscript, a supplementary table was added and a revised text was included in the methods section (lines 114-116): “Diet and physical activity were monitored as previously described during the washout period[30]. No significant differences in diet and physical activity were observed between intervention orders (Supplementary Table 1).”
Comment 6. The use of pupillometry is interesting, but the validation of this method as a direct indicator of ANS responses in a dietary context is not discussed in depth.
Response 6: Thank you for your valuable comment, partially answered with comment 2. Regarding dietary context, we have highlighted previous studies where pupillometry has been used to evaluate dietary impact on ANS. The revised text in the introduction, lines 85-86, now reads: “Additionally, pupillometry has been applied to assess the dietary impact on ANS responses[14], and its association with airway hyperresponsiveness has also been explored[29]. Given its practicality, pupillometry provides a unique opportunity to link dietary-induced ANS changes to physiological outcomes like lung function and airway inflammation.”
Comment 7. Heart rate variability data could have complemented the findings. It would be interesting if the authors included this information and discussion in the text.
Response 7. Thank you for this suggestion. We agree that the inclusion and discussion of heart rate variability (HRV) in this trial could provide complementary data and strengthen our findings. To address this, we have expanded the discussion to include the following text (lines 473-479): “Heart rate variability (HRV), an established marker of ANS function[11], could provide complementary insights into our pupillometry results. HRV reflects not only the balance of ANS response but also the impact of food on ANS parameters, particularly for short-term HRV measurements[63]. It has been closely associated with bronchial hyperresponsiveness[29]. However, the combined use of HRV and pupillometry to assess the effects of diet or meal interventions remains unexplored, and further standardization of both methods is needed[63].”
Comment 8. The absence of significant differences in FeNO between meals partially contradicts the initial hypothesis. Could the authors discuss this result in more detail, including possible mechanisms involved in FeNO regulation. For example, could the production of postprandial inflammatory mediators or changes in the microbiota play a relevant role?
Response 8. Thank you for this insightful comment. We agree that the absence of significant differences in FeNO between meals warrants further discussion. While the primary explanation for this result may be related to the limited sample size, additional factors could also contribute. Airway inflammation is regulated by different mechanisms, namely, genetic, environmental, and possibly by changes on microbiota. To address this, we have expanded the discussion section (lines 430-449) as follows: “The absence of significant differences may be partially explained by the limited sample size, which reduced the power to detect small effect sizes. Mechanistically, post-prandial inflammatory responses could play a role. High-fat meals have been associated with postprandial systemic inflammation, marked by increased levels of IL-6, TNF-α, and triglyceride [24, 70], which have been correlated with increased airway FeNO levels[70]. Neutrophilic airway inflammation patterns have also been linked to high-fat meals and it also suppresses bronchodilator recovery in asthma participants[32]. Additionally, the gut-lung axis may influence FeNO regulation, as dietary impacts on the gut microbiota can affect systemic inflammation [48] However, the acute nature of this study may have limited the potential for significant microbiota-ta-mediated effects. Greater vegetable diversity has been associated with lower airway inflammation[18], likely due to the antioxidant properties of these foods[48]. Dietary patterns rich in animal proteins and carbohydrates have been linked to increased asthma symptoms and reduced in lung function[71]. While the two tested meals differed in micronutrient composition, their comparable fat content could have minimized differences in postprandial airway inflammation and in lung function. Previous studies have evaluated specific interventions in meal-induced airway inflammation, such as avoiding exercise-induced airway inflammation [8] or supplementing meals with fibers and probiotics [9], which may have enhanced their ability to detect differences.”
Comment 9. The impact of specific micronutrients, such as polyphenols and omega-3 fatty acids, on the ANS is discussed, but the interaction of these compounds with inflammatory pathways could be further explored and better discussed.
Response 9. Thank you for this suggestion. We have included in the discussion a paragraph that further explored how polyphenols and omega-3 fatty acids interact with inflammatory pathways and potentially influence the autonomic nervous system (ANS) in lines 393-404: “Polyphenols modulate oxidative stress pathways by scavenging reactive oxygen species (ROS) and nitrogen species, thereby reducing LDL oxidation [54]. They also promote anti-inflammatory pathways by inactivating nuclear factor kappa B (NF-κB), down-regulating pro-inflammatory cytokines such as TNF-α and IL-6, and reducing the release of arachidonic acid, prostaglandins, and leukotrienes via the arachidonic acid signaling pathway [55]. Similarly, omega-3 polyunsaturated fatty acids (n-3 PUFA) exhibit anti-inflammatory effects by reducing levels of CRP, IL-6, and TNF-α, primarily through the inhibition of NF-κB activation and oxidative stress [56]. Furthermore, previous studies have shown that patients with low plasma levels of n-3 PUFA display an unbalanced pro-sympathetic response[57]. As oxidative stress promotes sympatho-excitatory effects, using antioxidants and anti-inflammatory agents, such as polyphenols and n-3 PUFA, may help mitigate adrenergic overdrive and restore autonomic balance [54].”
Comment 10. Demographic homogeneity (94% Caucasian) reduces the generalizability of the results. What can the authors say about this?
Response 10. Thank you for this important observation. We agree that the demographic homogeneity of our study population, with 94% of participants identifying as Caucasian, is a limitation that reduces the generalizability of our findings to more diverse populations. The recruitment strategy was based on convenience sampling in our region, which primarily consists of Caucasian individuals, and this may have contributed to the lack of diversity. Nevertheless, our study provides valuable insights into the effects of meal composition on autonomic nervous system responses, airway inflammation, and lung function, which serve as a foundation for future research. To address this, we have added the following statement to the discussion section (lines 482-488): “The demographic homogeneity of our study population, with 94% of participants identifying as Caucasian, limits the generalizability of our findings to more diverse populations. Previous studies have reported race-based differences in autonomic cardiovascular regulation in African Americans, which may influence autonomic responses to interventions [70] Future research should aim to include more diverse populations to validate these findings and explore potential racial or ethnic differences in autonomic nervous system regulation.”
Comment 11. The regression analysis adjusted for variables such as obesity and smoking is adequate, but an interaction analysis exploring the influence of these factors is missing.
Response 11: Thank you for this valuable comment. We appreciate the suggestion to explore interaction effects for variables such as obesity and smoking. Interaction analyses were conducted during the regression analysis to assess the potential influence of these factors. However, the interactions were not statistically significant and did not improved the model, therefore they were excluded from the final model. We included the following clarification in the methods section (lines 245-246): “Interaction effects for variables such as BMI and smoking status were explored but found non-significant; therefore, they were not included in the final model.”
Comment 12. The authors could expand the discussion on the biological mechanisms that link diet and ANS, including the role of the gut-brain axis and inflammatory mediators such as TNF-α and IL-6.
Response 12: Thank you for this valuable suggestion. We agree that the biological mechanisms linking diet and the autonomic nervous system, particularly the role of gut-brain axis and inflammatory mediators such as TNF alfa and IL-6, warrant further discussion. To address the specific mechanism we included in the discussion section the following paragraph (lines 381-390): “The interaction between diet and the autonomic nervous system (ANS) involves multiple biological pathways, including the gut-brain axis[8]. The vagus nerve plays a central role in mediating bidirectional communication between the gut microbiota and the brain[7]. Through the cholinergic anti-inflammatory pathway, the vagus nerve regulates systemic inflammation by suppressing pro-inflammatory cytokines, such as TNF-α [7]. Diet has a significant influence on vagal activity; for example, high-fat and high-carbohydrate diets have been shown to impair vagus nerve function in mouse models[53]. Additionally, inflammatory mediators such as IL-6, produced by enteric neurons, regulate the number and phenotype of microbe-responsive regulatory T cells in the gut, further linking diet, inflammation, and ANS activity[54].”
Furthermore, in the details regarding the specific impact of polyphenols and omega-3 fatty acids we further detailed the potential inflammatory mechanism, as specified in comment 9.
Comment 13. The authors did some more detailed subgroup analysis, highlighting differences between asthmatics, obese individuals, and non-obese individuals.
Response 13: Thank you for this observation. While our study included participants with asthma and obesity, the sample sizes within these subgroups were insufficient to perform detailed subgroup analyses with statistical power. As noted in the discussion, "This study focused on young, healthy adults; although we included some participants with asthma and obesity, their numbers were insufficient for subgroup analysis." However, to account for potential confounding effects, we included asthma and obesity classification as covariates in our statistical models. This allowed us to control their influence on the primary outcomes without performing separate subgroup analyses. We believe this approach strengthens the validity of our findings while acknowledging the limitation of small subgroup sizes.
Comment 14. It would be very interesting if the authors discussed in more detail the possible cumulative effects of repeated meals on ANS and lung function.
Response 14: Thank you for this suggestion. We agree that understanding the cumulative effects of repeated meals on ANS function and lung function is an important area for future investigation. Currently, most studies focus on long-term dietary interventions rather than the repetitive effects of individual meals. It would be valuable to explore how repeated meal exposures impact the progressive changes in lung function and autonomic nervous system parameters. To address this, we have expanded the discussion section as follows (line 510-516): “Long-term adherence to nutritionally rich plant-centered diet has been associated with slower lung function decline in a longitudinal cohort study[3]. These diets may exert cumulative benefits through sustained anti-inflammatory and antioxidant effects, which can influence systemic inflammation and oxidative stress over time. Additionally, a previous systematic review, highlighted that the duration of the dietary interventions influenced weight change and autonomic nervous system function[11], suggesting that repeated dietary exposure might promote a progressive modulation of autonomic nervous system.”
Comment 15. The authors could include a brief description of the societal impact of the study and the implications of the findings for the population studied.
Response 15: Thank you for these insightful comments. We have included in the conclusion section a brief description of the potential population, societal and also clinical impact of this study (line 538-545): “This study highlights the importance of daily dietary choices, demonstrating that meal quality acutely influences autonomic nervous system responses and airway inflammation. From a societal perspective, these results emphasize the need for policies that promote easy access to healthier dietary options, particularly in environments where fast-food consumption is prevalent. Clinically, the study suggests that dietary interventions, emphasizing balanced meals rich in unsaturated fats and antioxidants, could support parasympathetic activation and mitigate sympathetic overdrive, which is associated with obesity-related complications and respiratory conditions such as asthma[47]”
Comment 16. Avoid generalized statements about the benefits of Mediterranean diets. In this sense, the limitations of the study can be improved.
Response 16: Thank you for this important comment. We agree that generalized statements about the benefits of Mediterranean diets should be avoided, and we have revised the manuscript to ensure a more precise focus on the specific characteristics of the meals tested in this study. The following statement was added to the discussion section (line 464-465): “. The findings of our study should not be generalized to broader claims about the Mediterranean diet. Our meals differed primarily in micronutrient composition, which aligns with typical dietary habits but may yield different results than meals with more significant macronutrient variations, especially in fat and antioxidants content.”
Comment 17. It would be interesting to hear from the authors the importance of longitudinal studies on the subject of this study, if any.
Response 17: Thank you for this comment. We agree that longitudinal studies are crucial for understanding the long-term effects of diet on autonomic nervous system (ANS) function and lung function. We have addressed this topic in response to a previous comment (Comment 14), where we expanded the discussion to highlight the potential cumulative effects of repeated meals and the importance of long-term dietary interventions. Specifically, we emphasized that sustained adherence to nutrient-rich diets has been associated with slower lung function decline and may progressively modulate ANS parameters through anti-inflammatory and antioxidant mechanisms. This revised discussion can be found in lines 510-516 of the manuscript: “Long-term adherence to nutritionally rich plant-centered diet has been associated with slower lung function decline in a longitudinal cohort study[3]. These diets may exert cumulative benefits through sustained anti-inflammatory and antioxidant effects, which can influence systemic inflammation and oxidative stress over time. Additionally, a previous systematic review, highlighted that the duration of the dietary interventions influenced weight change and autonomic nervous system function[11], suggesting that repeated dietary exposure might promote a progressive modulation of autonomic nervous system.”
References
- Silva, D., et al., Meal-exercise challenge and physical activity reduction impact on immunity and inflammation (MERIIT trial). Contemp Clin Trials Commun, 2018. 10: p. 177-189.

Reviewer 2 Report
Comments and Suggestions for Authors
Dear Corresponding Author, thank you for submiting your article to Nutrients journal and congratulations on your work.
Brief Summary
Your study examines the acute effects of two isocaloric but micronutritionally different meals (Mediterranean vs fast food) on the autonomic nervous system, lung function and airway inflammation. Using a randomized crossover design with 46 participants, you have demonstrated significant differences in autonomic response between meals, with the Mediterranean meal promoting parasympathetic activity and fast food promoting sympathetic activity.
General Comments
I believe the work has some minor criticalities in some parts and I try to indicate the main areas for improvement which, in my opinion, can help enhance it:
- The discussion of mechanisms linking meal composition to autonomic response could be expanded
- A more detailed analysis of the clinical implications of the results would be useful, especially for asthmatic subjects
- The study limitations section could benefit from a more in-depth discussion on the generalizability of the results
Specific Comments
- Line 81: the BMI range is unclear because you indicate overweight/obese but specify only a generic ≥25
- Line 132: The rationale for choosing the measurement time (3 hours post-meal) should be better justified with literature references
- Table 1 could be improved in its graphical representation, while Table 2 is excessively extended to the left, beyond the margins of the Nutrients journal template
- Table 2: I would suggest adding a column showing the percentage differences between the two meals to facilitate interpretation
- "Methods" section: It would be useful to specify whether factors such as menstrual cycle were considered in female participants
These improvements will further strengthen an article that already has high scientific quality but can probably still be improved. If the authors will be able to implement or justify the points I have highlighted, I will give a definitive evaluation in second reading.
Author Response
Response to Reviewer #2:
Dear Corresponding Author, thank you for submiting your article to Nutrients journal and congratulations on your work.
Brief Summary
Your study examines the acute effects of two isocaloric but micronutritionally different meals (Mediterranean vs fast food) on the autonomic nervous system, lung function and airway inflammation. Using a randomized crossover design with 46 participants, you have demonstrated significant differences in autonomic response between meals, with the Mediterranean meal promoting parasympathetic activity and fast food promoting sympathetic activity.
General Comments
I believe the work has some minor criticalities in some parts and I try to indicate the main areas for improvement which, in my opinion, can help enhance it:
General comment 1: The discussion of mechanisms linking meal composition to autonomic response could be expanded
Response to General comment 1: Thank you for this valuable comment. We agree that a more detailed discussion of the mechanisms linking meal composition to autonomic nervous system responses could enrich the manuscript. To address this, we have expanded the discussion section to highlight key pathways and biological interactions involved. Specifically, we now elaborate on the role of inflammatory mediators (e.g., TNF-α, IL-6), oxidative stress, and the gut-brain axis in mediating these effects. The revised discussion emphasizes the following points: the gut-brain axis likely plays a central role, as dietary components influence gut microbiota composition and the production of short-chain fatty acids (SCFAs), which can modulate vagal signaling and systemic inflammation; we specified as an example the polyphenols and omega-3 fatty acids, which are present in the Mediterranean-like meal and may exert anti-inflammatory effects by reducing levels of TNF-α and IL-6 through the inhibition of NF-κB activation, as well as by mitigating oxidative stress. These effects may support parasympathetic activity and attenuate sympathetic overactivation; we also discuss some previous data regarding the effect of high-fat and high-carbohydrate meals, such as those typical of fast food, that have been shown to impair vagal function and promote systemic inflammation, thereby favoring sympathetic activation.
These additions were added to the discussion section between lines 381 to 407 : “The interaction between diet and the autonomic nervous system (ANS) involves mul-tiple biological pathways, including the gut-brain axis[8]. The vagus nerve plays a central role in mediating bidirectional communication between the gut microbiota and the brain[7]. Through the cholinergic anti-inflammatory pathway, the vagus nerve regulates systemic inflammation by suppressing pro-inflammatory cytokines, such as TNF-α [7]. Diet has a significant influence on vagal activity; for example high fat and high carbohydrate diets have been shown to impair vagus nerve function in mouse models[53]. Additionally, inflammatory mediators such as IL-6, produced by enteric neurons, regulate the number and phenotype of microbe-responsive regulatory T cells in the gut, further linking diet, inflammation, and ANS activity[54]. Nutrient content of a diet, including vitamins, proteins, PUFA, bioflavonoids, carotenoids, and other antioxidant metabolites, contributes to a long-term anti-inflammatory effect [55]. Poly-phenols modulate oxidative stress pathways by scavenging reactive oxygen species (ROS) and nitrogen species, thereby reducing LDL oxidation [56]. They also promote anti-inflammatory pathways by inactivating nuclear factor kappa B (NF-κB), down-regulating pro-inflammatory cytokines such as TNF-α and IL-6, and reducing the release of arachidonic acid, prostaglandins, and leukotrienes via the arachidonic acid signaling pathway [57]. Similarly, omega-3 polyunsaturated fatty acids (n-3 PUFA) exhibit anti-inflammatory effects by reducing levels of CRP, IL-6, and TNF-α, primarily through the inhibition of NF-κB activation and oxidative stress [58]. Furthermore, previous studies have shown that patients with low plasma levels of n-3 PUFA display an unbalanced pro-sympathetic response[59]. As oxidative stress pro-motes sympatho-excitatory effects, the use of antioxidants and anti-inflammatory agents, such as polyphenols and n-3 PUFA, may help mitigate adrenergic overdrive and restore autonomic balance [56]. A typical MdM, rich in these compounds, contributed to vagal activation, promoting a parasympathetic state, as evidenced by decreases in ACV and MCV [60].”
General comment 2: A more detailed analysis of the clinical implications of the results would be useful, especially for asthmatic subjects
Response to General comment 2: Thank you for this insightful comment. We agree that a more detailed analysis of the clinical implications of our findings, particularly for asthmatic subjects, is essential. To address this, we have first discussed the mechanisms associated with diet and asthma, particularly its impact on airway inflammation and then associated specific dietary patterns and our dietary intervention with asthma outcomes. These additions were included to the discussion section, lines 432 to 449.
General comment 3: The study limitations section could benefit from a more in-depth discussion on the generalizability of the results
Response to General comment 3: Thank you for this important observation. We acknowledge that the generalizability of our findings is limited by certain factors, which we have been now discussed further in the limitations section. Specifically, this study focused on young, healthy adults, with 94% of participants identifying as Caucasian, which limits applicability to more diverse populations (lines 483 to 486). While participants with asthma and obesity were included, their numbers were insufficient for subgroup analysis (lines 459-464). Additionally, as our meals differed primarily in micronutrient composition, the findings should not be generalized to broader claims about the Mediterranean diet or to meals with greater macronutrient variations (lines 464-465). The duration of the intervention and the effect of repeated dietary exposures and the potential impact of menstrual cycle was also recognized and discussed (lines 497-516).
Specific Comments
Specific Comments 1: Line 81: the BMI range is unclear because you indicate overweight/obese but specify only a generic ≥25
Response to specific comment 1: Thank you for highlighting this point. We have revised the manuscript to clearly define the BMI ranges for overweight (BMI 25–29.9 kg/m²) and obese (BMI ≥30 kg/m²) participants. This clarification is now in line 92-96.
Specific Comments 2: Line 132: The rationale for choosing the measurement time (3 hours post-meal) should be better justified with literature references
Response to specific comment 2: Thank you for this comment. Outcome measurements were obtained immediately before and 3 hours after each meal intake based on evidence from previous studies demonstrating that the peak effects of dietary interventions on autonomic nervous system responses and inflammatory markers occur within this timeframe. This approach allowed us to capture the acute postprandial changes most relevant to our outcomes. We have added this justification to the methods section and referenced supporting literature. Included in the methods section 2.3 Outcomes assessment (line 178-179) “Outcome measurements were obtained immediately before and 3 hours after each meal intake, as previous studies have shown that autonomic nervous system responses and postprandial inflammatory changes typically peak within this timeframe[32, 34].”
Specific Comments 3 and 4: Table 1 could be improved in its graphical representation, while Table 2 is excessively extended to the left, beyond the margins of the Nutrients journal template and Table 2: I would suggest adding a column showing the percentage differences between the two meals to facilitate interpretation
Response to specific Comments 3 and 4: Thank you for these valuable suggestions. Table 1 was graphically improved. Regarding Table 2 suggestion to include a column showing the percentage differences between the two meals. While we considered this approach, we found that percentage difference calculations were difficult to calculate in our dataset due to the nature of the variables and the presence of values near or equal to zero. Percentage differences rely on a denominator, and small or zero values can lead to instability, undefined results, or distorted percentages, which may misrepresent the true magnitude of differences between meals. To provide a more accurate and robust representation of the data, we have instead used the mean difference (or median difference, depending on the data distribution). This approach reflects more accurately the magnitude and direction of changes between the two meals and also aligns with the statistical tests performed, ensuring consistency and reliability in interpretation.
Specific Comment 5: "Methods" section: It would be useful to specify whether factors such as menstrual cycle were considered in female participants
Response to specific Comment 5: Thank you for your comment. We acknowledge that menstrual cycle phase may influence autonomic nervous system responses and could potentially modulate the effects of dietary interventions. As this information was not systematically asked during intervention we decided to include it as a limitation in the discussion section (line 492-496) “Although 57% of our participants were women, we did not monitor or control for menstrual cycle phase, which may influence autonomic nervous system responses. A previous study suggested that certain phases of the menstrual cycle might be associated with lower parasympathetic and higher sympathetic activity following a high energy meal. This factor could potentially modulate the autonomic response to dietary interventions and should be monitored[52].”
These improvements will further strengthen an article that already has high scientific quality but can probably still be improved. If the authors will be able to implement or justify the points I have highlighted, I will give a definitive evaluation in second reading.
We believe that all comments have been addressed. Nevertheless, we are happy to provide any further clarifications or make adjustments if needed.
References
- Silva, D., et al., Meal-exercise challenge and physical activity reduction impact on immunity and inflammation (MERIIT trial). Contemp Clin Trials Commun, 2018. 10: p. 177-189.

Round 2
Reviewer 1 Report
Comments and Suggestions for Authors
The authors made changes to the text and improved the manuscript significantly. The reviewer's recommendations were accepted by the authors, resulting in a more qualified, relevant and motivating manuscript for readers. I recommend that the manuscript be accepted for publication in its current format.